# Indefinite Graphene Nanocavities with Ultra-Compressed Mode Volumes

**DOI:** 10.3390/nano12224004

**Published:** 2022-11-14

**Authors:** Chunchao Wen, Zongyang Wang, Jipeng Xu, Wei Xu, Wei Liu, Zhihong Zhu, Jianfa Zhang, Shiqiao Qin

**Affiliations:** 1College of Advanced Interdisciplinary Studies, National University of Defense Technology, Changsha 410073, China; 2Hunan Provincial Key Laboratory of Novel Nano-Optoelectronic Information Materials and Devices, Changsha 410073, China

**Keywords:** graphene indefinite cavities, anomalous scaling rules, hyperbolic medium metamaterial, mode volumes

## Abstract

Explorations of indefinite nanocavities have attracted surging interest in the past few years as such cavities enable light confinement to exceptionally small dimensions, relying on the hyperbolic dispersion of their consisting medium. Here, we propose and study indefinite graphene nanocavities, which support ultra-compressed mode volumes with confinement factors up to $10^{9}$. Moreover, the nanocavities we propose manifest anomalous scaling laws of resonances and can be effectively excited from the far field. The indefinite graphene cavities, based on low dimensional materials, present a novel rout to squeeze light down to the nanoscale, rendering a more versatile platform for investigations into ultra-strong light–matter interactions at mid-infrared to terahertz spectral ranges.

## 1. Introduction

Optical nanocavities provide a indispensable platform to study various sorts of light–matter interactions, including light emission [1,2,3], optical nonlinearity [4,5,6], optomechanics [7,8], and quantum effects [9,10]. Surface plasmonic modes can be employed to shrink optical cavities down to the subwavelength scale, and in particular, graphene plasmons have accessible simultaneous extraordinary confinement and flexible tenability, covering the spectral regimes from the the mid-infrared to terahertz (THz) ranges [11,12]. When a graphene sheet is placed close to a metal surface, it supports a special type of highly confined and low-loss electromagnetic mode called acoustic graphene plasmons [13,14], based on which an acoustic graphene plasmon nanocavity with ultra-compressed mode volumes can be achieved [15,16]. Resonant metastructures in the excitation of ultrasharp states with small mode volumes, such as toroidal resonances, surface lattice resonances, and bound states in the continuum, have also been mentioned in previous work. [17,18]

Recently, hyperbolic metamaterials (HMMs) and metasurfaces [19,20,21,22,23] have attracted much research interest. Hyperbolic media have different signs of the principal elements of the permittivity or permeability tensors, which can be exploited for applications concerning the enhancement of the optical density of states, heat-transfer engineering, nonlinear effects, optical forces, superlensing [24,25,26,27,28,29,30], optical topological transition [31,32,33], etc. [34]. The open and extended hyperbolic dispersion curves or surfaces [35] accommodate propagating waves with huge wave vectors that are essential for optical cavity miniaturization [36,37,38]. To be specific, hyperbolic dispersions have been achieved in nanowire arrays [39,40] and layered metal-dielectric structures [41,42], based on which indefinite optical cavities have been obtained from the visible to near-infrared spectral regions. Besides metallic structures, HMMs based on 2D material such as graphene [43,44,45] in mid-infrared and terahertz ranges have also been proposed. Nevertheless, the possibilities of constructing indefinite nanocavities by relying on graphene-based HMMs and their contrasting optical properties have not been sufficiently explored.

In this article, we demonstrated a novel type of graphene indefinite nanocavity consisting of alternating graphene and silicon layers. The hyperbolic dispersion of such graphene-silicon HMM allows for propagating waves with large wave vectors and high effective indexes. This leads to ultra-confined modes with volume confinement factors up to $10^{9}$. Moreover, such modes show anomalous scaling rules compared to conventional optical cavities. The indefinite graphene nanocavity can be efficiently excited by far-field illuminations over a broadband range, and its capability to confine light into tiny dimensions can play a significant role in infrared spectroscopy [46], biosensing [47,48], and other applications [49,50], over the spectral regime from mid-infrared to THz.

## 2. Theoretical Model

The scheme for an indefinite graphene nanocavity is presented in Figure 1a, which consists of layered graphene-silicon HMMs. Figure 1b represents the 2D cavity array. In both scenarios, the thickness of single-layer dielectric silicon is *d* = 9 nm and its relative permittivity is set to be $\epsilon_{d}$ = 11.56. The graphene can be effectively treated as a surface current sheet characterized by its surface electric conductivity σ: $\sigma =\sigma_{inter}+\sigma_{intra}$, where $\sigma_{inter}$ and $\sigma_{intra}$ denote contributions from the inter-band and intra-band transition of electrons, respectively [51]:(1)$$\sigma_{intra\;}=\frac{2ie^{2}K_{B}T}{\pi \hbar^{2}\left(\omega +i/\tau \right)}\ln\left(2\cosh(\frac{E_{F}}{2K_{B}T})\right)$$
(2)$$\sigma_{inter}=\frac{e^{2}}{4}\left(\frac{1}{2}+\frac{1}{\pi }\arctan\frac{\left(\hbar \omega -2E_{F}\right)}{2K_{B}T}-\frac{i}{2\pi }\ln\frac{{\left(\hbar \omega +2E_{F}\right)}^{2}}{{\left(\hbar \omega -2E_{F}\right)}^{2}+{\left(2K_{B}T\right)}^{2}}\right)$$
where *T* is temperature; $K_{B}$ is Boltzmann constant; *e* is elementary electric charge; and *ℏ* is the reduced Planck constant. In this research, *T* = 300 K, $\tau =\mu E_{F}/eV_{F}^{2}$, $V_{F}=c/300$ is the Femi velocity, the carrier mobility of graphene μ = $10^{4}$
${\mathrm{cm}}^{2}$(vs)${}^{-1}$, and the chemical potential of doping graphene $E_{F}$ = 0.64 eV. Detailed simulation methods can be referred to Appendix A.

## 3. Results And Disscussion

### 3.1. Dispersion and Resonance Conditions

The iso-frequency contour of HMM is calculated from the following relation [52] at $f_{0}$ = 30 THz for transverse-magnetic polarized mode, which belongs to type II HMM [35]:(3)$$\cos\left(k_{z}d\right)=\cosh\left(\gamma d\right)-\frac{\gamma \beta }{2}\sinh\left(\gamma d\right)$$
where $\beta =-Z_{0}\sigma i/\epsilon_{d}k_{0}$, $Z_{0}=\sqrt{\mu_{0}/\epsilon_{0}}$ is the vacuum impedance and $\gamma =\sqrt{k_{x}^{2}-\epsilon_{d}k_{0}^{2}}$. This HMM is effectively isotropic on the transverse *x*-*y* plane [53]: $\epsilon_{x}=\epsilon_{y}=\epsilon_{d}+i\sigma Z_{0}/k_{0}d$ and $\epsilon_{z}=\epsilon_{d}$, indicating that the layered graphene-dielectric structure is effectively uniaxially anisotropic. The effective longitudinal permittivity along *z* ($\epsilon_{z}$) equals the permittivity of silicon and remains positive, while the transverse effective permittivity can be negative due to the material dispersion of the monolayer graphene. A conventional optical cavity is limited by the closed iso-frequency contours of the consisting medium. However, much larger wave vectors are allowed in this HMM, which sustain ultra-compressed mode-volumes of the indefinite optical cavity.

Cavity modes in the graphene indefinite cavity and the iso-frequency contour of HMM are both shown in Figure 2. Here, the red stars represent the resonant wave vectors of the graphene indefinite cavity (see Figure 3A–F) calculated through the Fabry–Perot resonant condition:(4)$$\delta \phi_{i}+Re\left(k_{i}\right)l_{i}=m_{i}\pi ,\;\;i=x,y,z,$$
where the $\delta \phi_{i}$ is the boundary phase shift; $k_{i}$ is the wave vector of mode; the integer $m_{i}$ represents the mode order; and $l_{x},l_{y}$, and $l_{z}$ are the length of a single cavity along the *x*, *y*, and *z*-directions, respectively. We assume that $l_{y}$ is infinite for 2D indefinite cavities (see Figure 1b) and $l_{x}=l_{y}$ for 3D indefinite cavities and confine our discussion to the special scenario of $k_{x}=k_{y}$.

### 3.2. Anomalous Scaling Rules of Graphene Indefinite Nanocavities

The spatial distributions of the electric field for a 2D indefinite cavity (see Figure 1b) are presented in Figure 3. It is clear that identical optical modes in these six cavities with different sizes ($l_{x}$, $l_{z}$) can be supported with the same resonant frequency and the same mode order ($m_{x}$, $m_{z}$) = (1, 1), and the confinement ability is comparable to that of acoustic graphene plasmon modes [15,54]. Indefinite graphene cavities with different sizes can resonate at a fixed frequency as long as the required resonant wave vectors are located on the the same iso-frequency contour. This means that the resonant wave vector can move along the iso-frequency curve as the size of the cavity scales down, which is not possible for conventional optical cavities. For example, the refractive indices ($n_{x}$, $n_{z}$) = ($k_{x}/k_{0}$, $k_{z}/k_{0}$) for a graphene indefinite cavity with the size combinations (62, 54), (44, 36), and (26, 18) nm are (80.6, 92.5), (113.6, 138.8), and (192.2, 277.6), respectively, all located on the iso-frequency curve. We can further shrink the dielectric thickness, making the effective mode index along *z* reach almost 300. However, the quantum effect should then be taken into consideration if the distance between neighbouring graphene sheets goes into the sub-nanometer regime.

Figure 4A–F shows the electric field distributions of the cavity modes with a fixed size [63, 54] nm for modes of different orders (2, $m_{z}$). In sharp contrast to conventional cavities, the higher-order mode resonates at lower frequencies, which are manifest in Figure 4. This is because the transverse and longitudinal components of the effective permittivity tensor ($\epsilon_{x}<$0 and $\epsilon_{z}>$0 ) of this bulk HMM have opposite signs. When the mode order $m_{z}$ increases, the larger resonant $k_{z}$ corresponds to a lower resonant frequency according to the following characteristic hyperbolic dispersion relation [44]:(5)$${k_{x}}^{2}/\epsilon_{z}+{k_{z}}^{2}/\epsilon_{x}=\omega^{2}/c^{2}$$

### 3.3. Far-Field Excitation of Indefinite Graphene Nanocavities

As a next step, we perform simulations of a 2D nanocavity array (see Figure 1b) with 50% filling ratio for different cavity sizes (i.e., the period of cavity array $p=2l_{x}$), which is illuminated by a far-field free-space plane wave. Figure 5 shows the transmission spectra for (1, 1) modes resonant at the same resonant frequency of 37THz for cavities of different sizes. As is also evident from Figure 5, for any cavity of a fixed size, the lower-order (1, 0) mode exhibits a higher resonant frequency than that of the higher-order (1, 1) mode, confirming the anomalous scaling rules discussed in the previous section.

### 3.4. Ultra-Compressed Mode Volumes of Graphene Indefinite Nanocavities

As a last step, we study the mode volume of graphene indefinite cavities by means of quasi-normal mode theory [15]. Figure 6 shows the obtained normalized mode volume of a 3D graphene indefinite cavity (red symbol and line), defined as $V_{ca}/\lambda_{0}^{3}$, where $V_{ca}=l_{x}l_{y}l_{z}$ (for the different vertical size $l_{z}$, we can calculate the vertical wave vector $k_{z}$ by the z-direction Fabry—Perot resonant condition by Formula (Equation 4). In the next step, the tangential wave vector $k_{x}$ and the tangential size $l_{x}$ of the small cavity can be obtained by the hyperbolic iso-frequency contour shown in Figure 2 and the x-direction cavity resonant condition. Because the effective permittivity is isotropic and unchanged along all tangential directions for a multilayer system, we can assume the $k_{y}$ = $k_{x}$ and $l_{x}$ = $l_{y}$ for a 3D cavity roughly characterizes the mode volume. A bowtie photonic crystal cavity with a recorded deep subwavelength mode confinement factor $10^{-5}$∼$10^{-4}$ [55] and similar metal-insulator-metal (MIM) indefinite cavities have beem experimentally demonstrated, and in the near-infrared range they have also been shown to obtain great field confinement [41]. Although the resonant wavelength of the graphene indefinite cavity (approximately 10 μm here, or longer wavelengths) is much larger than that of the MIM indefinite cavity (approximately 2 μm; refer to Ref. [41]), the obtained normalized mode volume of our graphene indefinite cavities can reach up to $10^{-9}$ (that is, a mode-volume confinement factor up to $10^{-9}$), which is approximately two orders of magnitude smaller than that of the indefinite MIM cavities). The normalized mode volumes achieved here are comparable to the recently reported acoustic graphene plasmon cavities [15]. The extraordinary confinement we have achieved mainly relies on the open hyperbolic dispersion curves, which can be employed to further squeeze light down to atomic scales [54].

## 4. Conclusions

In conclusion, we propose and demonstrate graphene indefinite nanocavities with ultra-compressed mode volumes and extraordinary optical confinement at mid-infrared and THz spectral regimes. The normalized mode volume can reach approximately $10^{-9}$, two orders of magnitude smaller than the widely studies MIM indefinite cavities. Our indefinite graphene nanocavities can be efficiently excited from the far field and manifest anomalous scaling laws of the resonances, which can function as a promising playground to study extreme light–matter interactions and explore tunable high-performance metadevices with desired functionalities.

## Figures and Tables

**Figure 1 nanomaterials-12-04004-f001:**
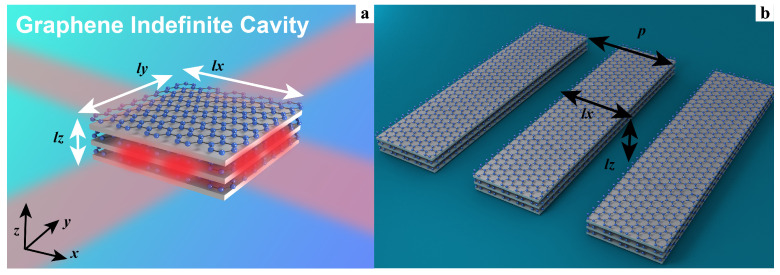
Schematic of the graphene indefinite nanocavity.(**a**). Perspective view of the a nanocavity made of mulilayered graphene-silicon HMM with indefinite permittivity. (**b**). 2D graphene indefinite cavity array.

**Figure 2 nanomaterials-12-04004-f002:**
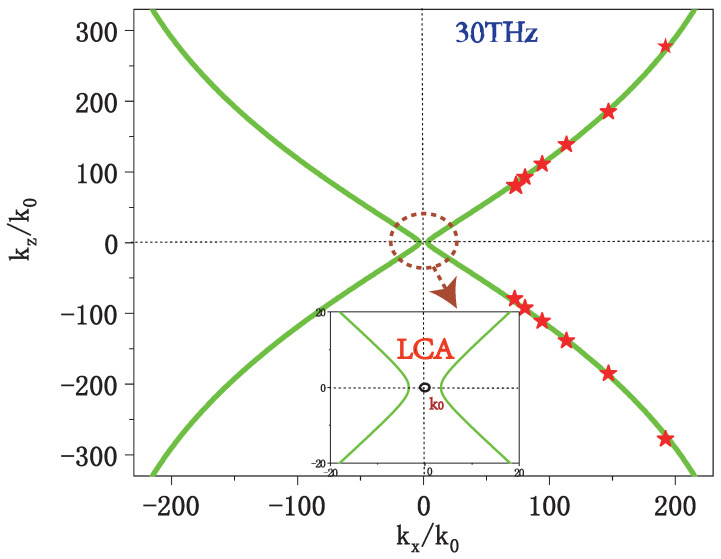
Iso-frequency contour of the graphene-silicon HMM in x-z plane at 30THz. Hyperbolic curves (green line) represent allowed propagating modes inside the multilayered metamaterial (calculated from Equation (Equation 3)), and the red stars denote resonant cavity modes. The black circle with radius $k_{0}$ around the origin represents a cross section of light cone in air (LCA).

**Figure 3 nanomaterials-12-04004-f003:**
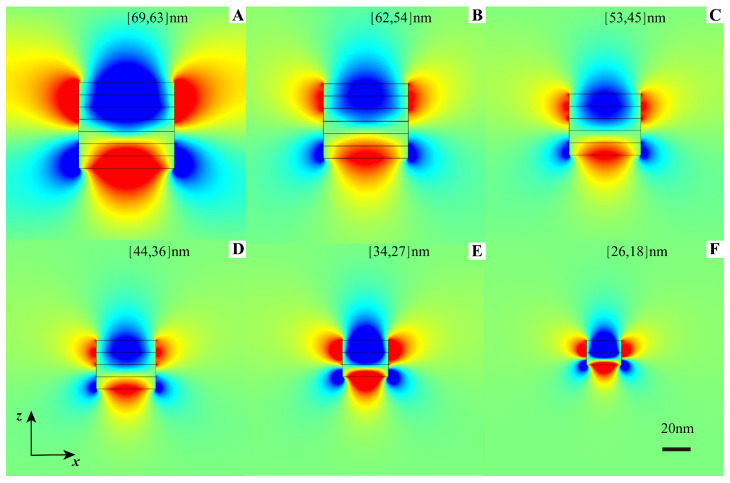
Mode distribution profiles with different cavity sizes. Electric field $E_{x}$ of the (1, 1) mode for graphene indefinite nanoscale cavities with different (width, height) sizes but at the same resonant frequency 30THz.

**Figure 4 nanomaterials-12-04004-f004:**
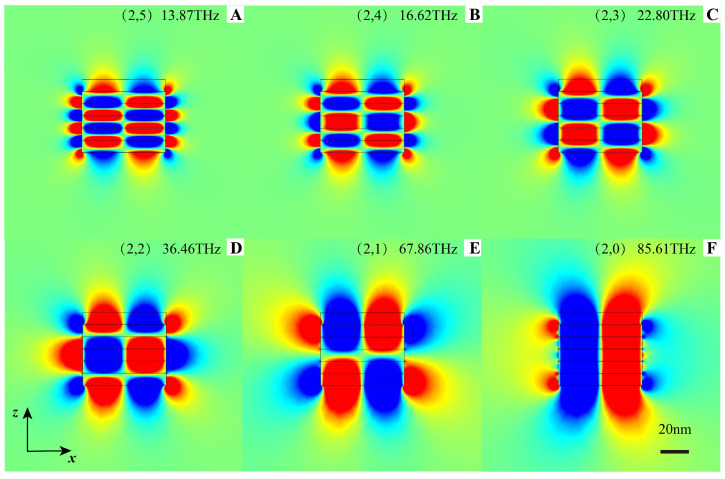
Resonate frequency with different mode order. Electric field $E_{x}$ of six cavity modes ($2,m_{z}$) with different resonate frequencies but at the same cavity size [62, 54] nm.

**Figure 5 nanomaterials-12-04004-f005:**
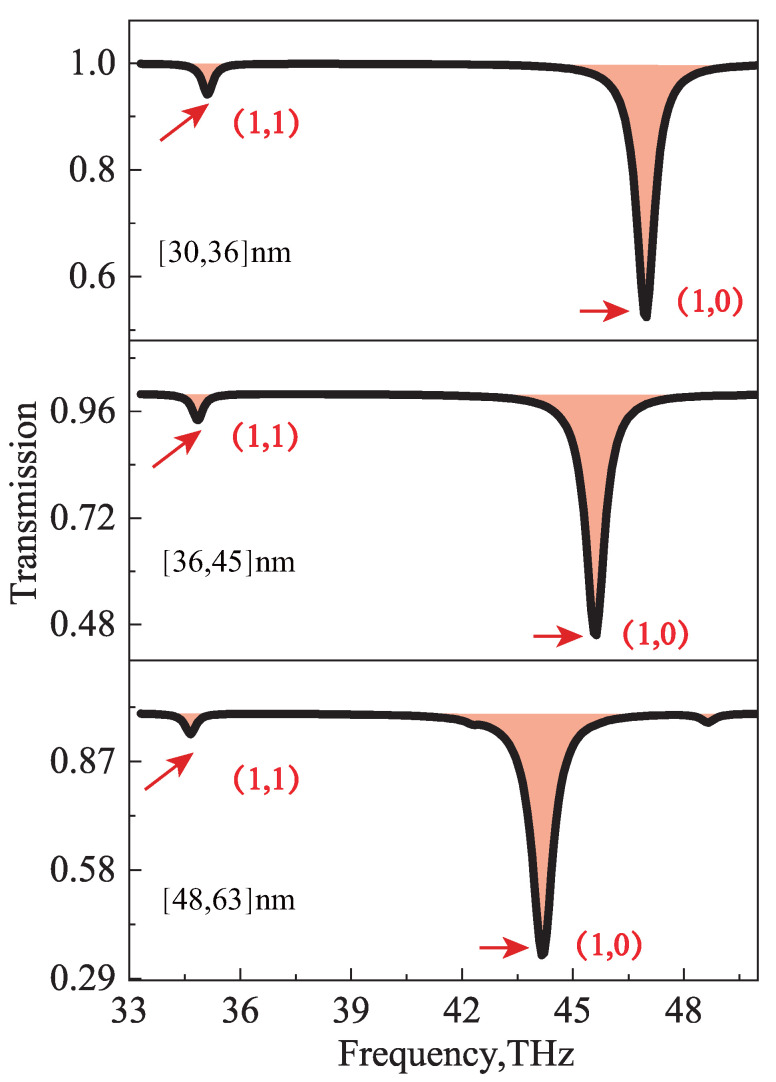
Transmission spectra of periodic indefinite nanoscale cavity array. 1D cavity array with a 50% cavity area filling ratio for different cavity sizes. Both higher−order (1, 1) and lower−order (1, 0) modes are indicated. All (1, 1) modes resonate at the same frequency of 37 THz, and lower order (1, 0) modes resonate at other higher higher frequencies.

**Figure 6 nanomaterials-12-04004-f006:**
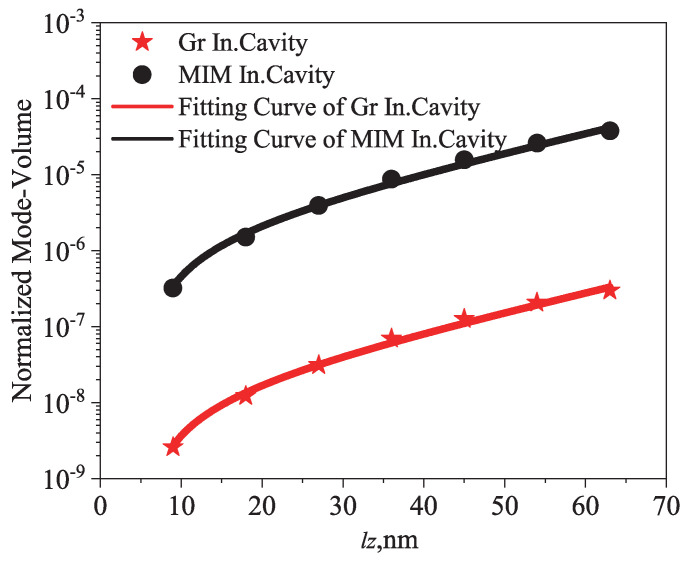
The obtained normalized mode−volume of indefinite cavities. Graphene indefinite cavities resonate at the far−infrared spectrum, where red symbols are calculated from the cavities in Figure 3, the red line is the fitting curve and MIM indefinite cavity resonates at the near−infrared range (black), calculated from reference [41], as function of the different length $l_{z}$ of a single indefinite nanoscale cavity.

## Data Availability

Relevant data is included in the manuscript.

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
