# Peer review of "Indefinite Graphene Nanocavities with Ultra-Compressed Mode Volumes"

_nanomaterials, 2022, doi:10.3390/nano12224004_

Round 1
Reviewer 1 Report
This study discusses the design and study of graphene nanocavities with hyperbolic properties that yield ultralow modal volumes along with robust confinement of electromagnetic fields. Extensive numerical studies have been conducted to extract the spectral properties and features of the devised metaplatform. The developed study possesses significant novelty, and the results look promising, however, there are some important points that must be addressed in the revised version of the draft before proceeding further. I listed my comments below:
1) Although the bibliography contains important information about the background of the context, the authors also should consider the use of resonant metastructures in the excitation of ultrasharp states with loa mode volumes, such toroidal resonances, surface lattice resonances, bound states in the continuum, etc. See: Laser & Photonics Reviews 14(11), 1900326 (2020), Advanced Optical Materials, 9(7), 2001520 (2021).
2) Although the obtained modal volume is interesting, the authors should quantitatively evaluate the performance of the system with analogous systems. I have seen much lower modal volumes in previous reports: Science Advances 4(8), eaat2355 (2018).
3) The mathematical technique behind the calculation of the model volume of the developed system?
4) The mode volume of the system was presented in a normalized fashion. Arbitrary results will help to understand how the field can be squeezed strongly?
5) What type of simulation method was employed to study the system? The details and settings of the analysis should be provided.
Reviewer 2 Report
The authors present a well executed study about anomalous scaling laws in hybrid nanocavities with very strong field confinements in the THz range. The work is well thought out and reasoned. Clear figures accompany the manuscript. I recommend publication as is.
Reviewer 3 Report
Dear Authors,
I have read your manuscript and your work is interesting and well described.
I suggest only to add a reference to understand from where formulas 1a and 1b are coming from.
Best Regards
Round 2
Reviewer 1 Report
The comments and concerns have been responded correctly, therefore, the work can be published as is.